# Efficacy Evaluation of 10-Hydroxy Chondrofoline and Tafenoquine against *Leishmania tropica* (HTD_7_)

**DOI:** 10.3390/ph15081005

**Published:** 2022-08-15

**Authors:** Sayyed Ibrahim Shah, Fazli Nasir, Nadia Shamshad Malik, Muhammad Alamzeb, Muhammad Abbas, Inayat Ur Rehman, Fazli Khuda, Yasir Shah, Khang Weh Goh, Alam Zeb, Long Chiau Ming

**Affiliations:** 1Department of Pharmacy, Abdul Wali Khan University Mardan, Khyber Pakhtunkhwa 23200, Pakistan; 2Department of Pharmacy, University of Peshawar, Khyber Pakhtunkhwa 25100, Pakistan; 3Faculty of Pharmacy, Capital University of Science and Technology, Islamabad 44000, Pakistan; 4Department of Chemistry, University of Kotli, Kotli 11100, Pakistan; 5Faculty of Data Science and Information Technology, INTI International University, Nilai 71800, Malaysia; 6Department of Biochemistry, University of Malakand, Khyber Pakhtunkhwa 18800, Pakistan; 7PAP Rashidah Sa’adatul Bolkiah Institute of Health Sciences, Universiti Brunei Darussalam, Gadong BE1410, Brunei

**Keywords:** cutaneous leishmaniasis, drug assay, parasitic disease, sand flies, BALB/c mice

## Abstract

Leishmaniasis is affirmed as a category one disease (most emerging and unmanageable) by the World Health Organization (WHO), affecting 98 countries with an annual global incidence of ~1.2 million cases. Options for chemotherapeutic treatment are limited due to drug resistance and cytotoxicity. Thus, the search for new chemical compounds is instantly desirable. In this study, we used two compounds, i.e., 10-hydroxy chondrofoline and tafenoquine, for their antileishmanial activity against *L. tropica* (HTD_7_). First, the cytotoxicity assay of the test compounds against THP-1 cells was carried out, and these compounds were found safe. Intra-THP-1 amastigote activity (*in vitro*) was performed, which was then followed by the *in vivo* activity of 10-hydroxy chondrofoline in the murine cutaneous leishmaniasis (CL) model. A total of three concentrations were used, i.e., 25, 50, and 100 µM, to check the *in vitro* activity of the test compounds against the amastigotes. 10-hydroxy chondrofoline was found to be the most potent compound *in vitro* (and thus was selected for *in vivo* studies) with an LD_50_ value of 43.80 µM after 48 h incubation, whilst tafenoquine had an LD_50_ value of 53.57 µM. *In vivo* activity was conducted by injecting 10-hydroxy chondrofoline in the left hind foot of the infected BALB/c mice, where it caused a statistically significant 58.3% (F = 14.18; *p* = 0.002) reduction in lesion size (0.70 ± 0.03 mm) when compared with negative control (1.2 ± 0.3 mm).

## 1. Introduction

The leishmaniases are a group of neglected tropical diseases transmitted through the bite of the female sand fly that typically affects populations in low- and middle-income countries, including Pakistan [1]. There are two types of *Leishmania* infection: visceral leishmaniasis (VL) and cutaneous leishmaniasis (CL), with CL being the most prevalent form of the disease [2,3].

According to World Health Organization, more than 80% of new CL cases happened in lower-income countries, including Pakistan, India, Libya, Afghanistan, Algeria, and Brazil [4]. CL is caused predominantly by the *L. major* and *L. tropica* species. This disease is characterized by the presence of lesions on the bite site whose appearance may range from the appearance of nodules to ulcerative lesions, which usually leave disfiguring scars after its healing. Initially, the lesion is in the form of a papule, which over time is converted into the nodule and then into an ulcer during the next 1–3 months. *Leishmania* ulcer has a crusted base and raised margins. The lesion can become painful due to secondary bacterial infection [5,6]. In the last decade, the global number of CL cases has increased due to inadequate vector reservoir control and increased opportunistic infections with AIDS. The migration of refugees from war-affected areas also has resulted in the spread of the disease to new, previously non-endemic parts [7,8]. The parasite strain used in this study was *L. tropica* (MHOM/AG/2015/HTD_7_), where MHOM represents *Homo sapiens*, AG for Afghanistan, 2015 the year strain was isolated whilst HTD represents Hospital for Tropical Diseases, and the subscript seven is the specific code for the patient from whom the strain was isolated [9].

Currently, the first choice for treating CL is antimonial compounds, i.e., meglumine antimoniate (Glucantime^®^) and sodium stibogluconate (Pentostam^®^). Apart from this, alternative treatment options include the use of amphotericin B, miltefosine, paromomycin, and pentamidine [10,11,12]. Usage of these drugs is associated with drawbacks including toxicity (especially hepatotoxicity), painful method of administration, i.e., intralesional injections, extended duration of the treatment regimen, and high cost of the treatment, all of these leading to a decrease in patient compliance, which alongside inadequate therapeutic response have led to the emergence of newer, more resistant strains of the parasite [13,14]. There is an urgent need for the development of new compounds that can treat the infections whilst having lesser side effects as compared to the existing treatment [15]. In this regard, various approaches can be used, which include the use of compounds (that are used for the treatment of other diseases) as well as finding new treatment options from nature, i.e., plants.

Tafenoquine (Figure 1) is an 8-amminoquinolone that is typically used for the treatment of Malaria. It has also shown good activity against other parasitic diseases such as Trypanosomiasis caused by *T. brucei* [16]. Moreover, its efficacy against VL caused by *L. donovani* has been reported. Tafenoquine works by weakening the overall bioenergetic metabolism of *Leishmania*, resulting in a quick fall in intracellular Adenosine Triphosphate (ATP) level with no effect on plasma membrane permeability. It also causes mitochondrial dysfunction by restraining the cytochrome-c reductase with a decline in the oxygen utilization rate and ultimately depolarization of mitochondrial membrane potential. This also results in Reactive Oxygen Species (ROS) generation, the rise of intracellular Ca^2+^ level, and associated nuclear DNA disintegration, leading to apoptosis [17]. Moreover, its antileishmanial activity is dependent upon various factors, e.g., the uptake of the drug, biochemistry of the cell membrane, etc. The uptake of the tafenoquine into the amastigotes of the parasite is a pH and temperature-dependent process. Moreover, it (the uptake) is also dependent upon the sterol content of the cell membrane, i.e., a decrease in the membrane sterol content will lead to a decreased uptake of the drug in the amastigotes, resulting in the decreased therapeutic efficacy of the drug [18].

*Berberis glaucocarpa* is a small shrub native to the Himalayan region and has been used in traditional medicine for the treatment of various diseases [19]. Antileishmanial activity of alkaloids is well reported [20]. They consist of carboxyl and acetate groups in their structure which are responsible for their structure-activity relationship (SAR). Moreover, the exact mechanism behind the antileishmanial activity of an alkaloid depends upon the exact structure and can range from iron chelation to the decreased uptake of the promastigotes by the dendric cells [21,22]. 10-hydroxy chondrofoline (Figure 2) is an alkaloid isolated from the root and stem bark of *Berberis glaucocarpa* (commonly found in northern Pakistan). It is a light-yellow powder, having the molecular formula of C_37_H_40_N_2_O_7,_ whilst the molecular weight of the compound is 622 Da, with a melting point (m.p) of 195–197 °C.

In this study, to find a potential new treatment with decreased side effect profile for CL, we have reported the *in vitro* activities of two compounds, i.e., 10-hydroxy chondrofoline and tafenoquine, followed by the *in vivo* activity of 10-hydroxy chondrofoline against *L. tropica* (HTD_7_).

## 2. Results

### 2.1. In Vitro Cytotoxicity of Test Compounds and Standard Drug against THP-1 Cells

Table 1 shows the percentage viability of the control and drug-treated THP-1 cells. None of the drugs showed significant toxicity when compared with the control group (Control THP-1 cells only) (97.1% ± 1.1). Of note, Fisher Scientific USA cell culture protocol indicated that the minimum untreated cell viability could be 95% for healthy log-phase cultures [23]. Here in our experiment, the untreated log-phase THP-1 cells were 97.1% viable, showing admirable health status. It was fascinating to know that our target compounds, 10-hydroxy chondrofoline and tafenoquine, showed minimum cytotoxicity against THP-1 cells.

### 2.2. In Vitro Evaluation of Antileishmanial Activity

In this study, we tested two compounds, i.e., 10-hydroxy chondrofoline and tafenoquine, against the intracellular amastigotes *in vitro*. *Leishmania* parasite has two stages in their life cycle, i.e., promastigotes and amastigotes [24]. The latter is the infective form and causes the infection of the invaded macrophages (inside the phagolysosome) [25]. This provided the rationale behind choosing the amastigotes for this assay.

Intracellular amastigotes of *L. tropica* clinical field isolate HTD_7_ were exposed to three different concentrations, i.e., 25, 50, and 100 µM at 37 °C for 24 and 48 h, respectively. Positive control was meglumine antimonate whilst the negative control was without any drug. Percent inhibition was calculated by counting intracellular amastigotes under a hemocytometer in comparison to untreated controls. LD_50_ values (the inhibitory concentration of compound that kills 50% of the *Leishmania* amastigotes) were calculated after dispensation of the percent inhibition value in GraphPad Prism-6 software. Giemsa stained THP-1 cells infected with *L. tropica* 48 h post-infection, and post-treatment are shown in Figure 3. Amastigotes can be seen in the cytoplasm of the THP-1 cells. Positive control (meglumine antimoniate) showed limited effectiveness, i.e., 18.7, 25.2, and 54.5% inhibition at 25, 50, and 100 µM after 48 h incubation (LD_50_ value 87.68 µM) against the *L. tropica* HTD_7_ Intra-THP-1 amastigotes as compared to other three compounds. 10-hydroxy chondrofoline was the most effective compound against amastigotes of *Leishmania tropic* HTD_7_ eliminating 67.9, 38, and 19.8% of the intra-THP-1 amastigotes at 100, 50, and 25 µM, with an LD_50_ value of 43.80 µM after 48 h incubation. Tafenoquine at 25, 50, and 100 µM inhibited14.4, 30.9, and 59.2% of the parasite load, respectively, with an LD_50_ value of 53.57 µM after 48 h (Figure 4).

### 2.3. Testing In Vivo Leishmanicidal Activity Using BALB/c Mice Experimentally Infected with Leishmania Tropica (HTD_7_)

Cutaneous lesions developed on the 62nd-day post-inoculation, characterized by swelling on the infected left foot pad of all the infected mice (Figure 5). Treatment was initiated after the development of lesions (63rd-day post-inoculation) and was carried out for 20 days. For this purpose, first, the size of the lesions was measured by using a vernier caliper. The mean diameter of the group one mice-infected footpad was 4 mm, with a lesion size of 1.2 ± 0.3 mm. Similarly, for group two, the mean size of the infected footpads was 4.4 mm, with a lesion size of 1.5 ± 0.4 mm, while for group three, the mean size of the infected footpads was 4.6 mm with a lesion size of ~1.55 ± 0.3 mm. Treatment comprised over a period of 20 days. The mice were daily observed for their health, activities, and lesion peripheral health status. Compounds, i.e., standard and the test compounds, were first dissolved in DMSO and then diluted with PBS with a final DMSO concentration of 1% *v*/*v*. Intralesional injections were given at a dose of 20 mg/kg with an injection volume of 12 µL. As compared to the diameter of the lesion of the negative control group (1.2 ± 0.3 mm), lesion sizes of meglumine antimonate and 10-hydroxy chondrofoline treatment recorded were 0.56 ± 0.07 mm and 0.70 ± 0.03 mm respectively. Although in the *in vitro* assay, 10-hydroxy chondrofoline was the most effective compound, the *in vivo* results were not as expected, i.e., meglumine antimoniate was more effective. This indicates a 53.4% reduction in lesion size with meglumine antimonate and a 41.7% decrease in lesion size after treatment with 10-hydroxy chondrofoline after 10 injections for 20 days treatment (Figure 6).

## 3. Discussion

For authorization of a drug, one of the most essential parameters is that the given compound should not be toxic to the host cells. For this purpose, we evaluated the cytotoxicity of our compounds and the standard drug on THP1 cells by using a trypan blue exclusion assay. The *in vitro* results exhibited 10-hydroxy chondrofoline to be the most potent drug as compared to meglumine antimoniate. The rationale behind the inadequate effectiveness of meglumine antimonate might be due to resistance developed by the *L. tropica* HTD_7_ strain as meglumine antimonate show different responses to different *Leishmania* species, e.g., Jain et al. reported IC_50_ values from 10 µg/mL (susceptible strain) to more than 128 µg/mL for the resistant strain of *L. panamensis.* This response of the resistance strain resembles our results; as mentioned before, meglumine antimonate is the normal therapy used in Afghanistan, so this strain may have developed resistance against meglumine antimonate [26]. Combination therapy is effective in the elimination of various pathogens. For example, Shakori et al. used meglumine antimonate alone and in an amalgamation with verapamil against *L. tropica*. They reported IC_50_ values of 225.14 µg/mL and 116 µg/mL for meglumine antimonate only and in combination with verapamil, respectively [27]. Similarly, tafenoquine also showed good activity, and our results are supported by several studies conducted on tafenoquine as antileishmanial against different species of *Leishmania* [28].

In vivo study was performed on mice. Rodents, especially mice, are commonly used as a model to study the *Leishmania* infection has been very useful in interpreting events happening through the inborn immune response and those concerned with the demarcation of infection [29]. The murine CL model used here has been reported previously [30], and thus, the reason for choosing this model for *in vivo* study. Although in the *in vitro* assay, 10-hydroxy chondrofoline was the most effective compound, the *in vivo* results were not as expected, i.e., meglumine antimoniate was more effective. The possible reason can be the sex difference in the mice, i.e., both male and female were used, and it is well documented that the sex difference (present in a test group) affects the susceptibility of the *Leishmania* towards the drugs, thus affecting the overall results [22]. Limitations of this study include the mixing of both genders, i.e., males and females in the *in vivo* study, which may have affected the susceptibility of the *Leishmania* parasite towards the drugs. Moreover, as a vernier caliper was used for measuring the size of CL lesions, it might have resulted in a false reading whilst measurement of the lesions due to the possibility of human errors

## 4. Materials and Methods

### 4.1. Materials

Test compounds 10-hydroxy chondrofoline and tafenoquine were kindly provided by Dr Muhammad Alamzeb (Assistant Professor, University of Kotli, Azad Jammu and Kashmir, Pakistan) and Dr Vaneesa Yardley (Assistant Professor, London School of Hygiene and Tropical Medicine (LSHTM), UK), respectively. Meglumine antimoniate was gifted by Prof. Simon Croft (LSHTM) and was used as such *L. tropica* (MHOM/AG/2015/HTD_7_) was kindly provided by the London School of Hygiene and Tropical Medicine (LSHTM), UK. BALB/c mice were provided by the Department of Pharmacy, University of Peshawar, Pakistan. All the solvents were of analytical grade and were used as received.

### 4.2. Methods

#### 4.2.1. Culture of *L. tropica* (HTD_7_)

*Leishmania tropica* were cultured in a 25 cm^2^ polystyrene Nunc tissue culture flask containing RPMI-1640 growth medium with 10% heat-inactivated Fetal Bovine Serum (HI-FBS) and 1% Penicillin-streptomycin solution added. Culturing was performed aseptically in a class-II biosafety cabinet. The culture flask was kept in an incubator at 26 °C under anaerobic conditions. The culture was regularly observed and counted daily for 7 days through Neubauer Hemocytometer. Upon the change of the colour, the medium was sub-passaged by spinning the culture in a centrifuge (Sigma, Saint Louis, MA, USA) for 10 min at 1200× *g*. The obtained pellet was then re-suspended in 1 mL of fresh medium, counted, and introduced into another culture flask containing 5 mL of fresh medium at 1 × 10^5^ promastigotes per mL. Unused promastigotes were cryopreserved by using cryo-medium (RPMI 1640 + 50% FBS + 7.5% DMSO) in liquid nitrogen at −196 °C. Figure 7 shows the growth curve for *L. tropica* HTD_7_ promastigotes over 8 days period. In the first 3 days of the culture, a little growth was detected. However, on day 4, the log phase of the growth was noticed, and the mid and late log phases were achieved on days 5 and 6. Peak growth was attained on day 6; the culture showed a stationary stage with a promastigote concentration of 1 × 10^7^ mL.

#### 4.2.2. Infection of THP-1 Macrophages with *L. tropica* (HTD_7_)

Preserved THP-1 human acute monocytic leukemia cell line was cultured in RPMI-1640 medium supplemented with 10% HI-FCS at 37 °C for at least 7 days, with 48–72 h medium change in 85% humidified CO_2_ incubator. To differentiate monocytes into macrophages, exponentially growing THP-1 cells were plated at 2 × 10^4^ cells/well in 100 µL of phorbol myristate acetate 20 ng/mL (PMA) for 48 h.

THP-1 cells differentiated into macrophages after treatment with PMA were infected with late-log or stationary promastigotes of *L. tropica* (HTD_7_) isolated at 10:1 parasite to macrophage proportion and incubated at 34 °C for 15 h. On the following day, the un-endocytosed promastigotes of *L. tropica* (HTD_7_) were washed off using a pre-warmed RPMI growth medium without fetal bovine serum and incubated for another 16–24 h at 34 °C to improve the infection.

#### 4.2.3. In Vitro Cytotoxicity of Test Compounds and Standard Drug against THP-1 Cells

For cytotoxicity assays, the THP-1 cells were incubated with 2 test compounds and a standard drug for 24 and 48 h and were compared against the negative control as per the given protocol. The effect of test compounds and the positive control on THP-1 macrophages were scrutinized using the trypan blue exclusion test to determine the viability of the cells. For this purpose, a hemocytometer was loaded with 20 µL of the cell suspension and examined immediately under a microscope at both low (10 × 10) and high (40×) magnification. Stained cells (blue in colour) were considered dead. Cell viability was calculated
% Viable THP-1 cells = [1.00 − (Number of blue THP-1 cells ÷ Number of total THP-1 cells)] × 100(1)

#### 4.2.4. In Vitro Evaluation of Antileishmanial Activity

The THP-1 macrophages were seeded on Lab Tek tissue culture slides (USA Scientific) at a density of 2 × 10^4^ cells/well. For better adherence, these cells were allowed for 2 h inside a CO_2_ incubator at 34 °C. The slides were then washed with the serum-free RPMI-1640 medium to remove any non-adherent THP-1 macrophages. They were infected with late-log or stationary promastigotes of *L. tropica* (HTD_7_) isolate at 10:1 parasite to macrophage proportion and incubated at 34 °C for 15 h. On the following day, the un-endocytosed promastigotes of *L. tropica* (HTD_7_) were washed off using a pre-warmed RPMI growth medium without fetal bovine serum and incubated for another 16–24 h at 34 °C to improve the infection. The infected macrophages were then incubated with 25, 50, and 100 µM of 10-hydroxy chondrofoline, tafenoquine, and meglumine antimoniate, respectively, in triplicate for 48 h at 34 °C. This was then followed by fixation by methanol for 1 min and then was stained by Giemsa stain. In each well, 100 macrophages were counted to estimate the LD_50._ As these compounds are water-insoluble, therefore, to solubilize, they were first dissolved in Dimethylsulfoxide (DMSO) at 1 mg/mL and were further diluted with the RPMI-1640 growth medium. The final volume of the DMSO was kept at 1% (*v*/*v*) to avoid the cytotoxicity caused by DMSO itself, as it has been reported that a concentration greater than 1% (*v*/*v*) may cause cytotoxicity [31] and hence can produce false results. Meglumine antimoniate was used as a positive control. The rationale behind selecting the meglumine antimoniate was that it is a first line antileishmanial drug, with numerous studies reporting its use as a positive control [32]. The negative control group consists of RPMI-1640 diluted with DMSO with a final concentration (DMSO) of 1% (*v*/*v*).

#### 4.2.5. Testing In Vivo Leishmanicidal Activity Using BALB/c Mice Experimentally Infected with *L. tropica* (HTD_7_)

Ethical approval for the animal study was obtained from the Department of Pharmacy, University of Peshawar, Pakistan, ethical committee (Ref. 11/EC/-16/Pharm). Based on the preliminary *in vitro* assay results, 10-hydroxy chondrofoline was selected for *in vivo* assay. The *in vivo* assay was performed using 8–12 weeks old 12 males and 6 female BALB/c mice, weighing 25 to 35 gm, initially bred in the animal house of the Department of Pharmacy, University of Peshawar, Pakistan. Mice were subcutaneously inoculated into the hindfoot with 1× 10^6^ of stationary phase *Leishmania tropica* HTD_7_ parasites (metacyclic promastigotes). A total of 18 mice were alienated into 3 groups, each containing 6 BALB/c.

Details are as:

**Group 1**: This was the negative control group having 4 males and 2 females. This group was injected with PBS (with 1% (*v*/*v*) DMSO) (pH 7.4) into the cutaneous lesions developed after inoculation of the metacyclic promastigotes.

**Group 2:** This was the positive control group consisting of 4 males and 2 females. This group was injected with a reference drug (meglumine antimoniate) dissolved in PBS (with 1% (*v*/*v*) DMSO) given at a dose of 20 mg/kg and with an injection volume of 12 µL into the cutaneous lesions.

**Group 3:** This was the experimental group consisting of 4 male and 2 female mice. This group was injected with test compound 10-hydroxy chondrofoline dissolved in PBS (with 1% (*v*/*v*) DMSO) given at a dose of 20 mg/kg and with an injection volume of 12 µL into the cutaneous lesions.

#### 4.2.6. Statistical Analysis

Statistical significance was determined by using a one-way analysis of variance (ANOVA), followed by Bonferroni correction using GraphPad Prism 6.0 software (GraphPad Software 2012; San Diego, CA, USA).

## 5. Conclusions

The novelty of this study is the murine CL model of *L. tropica* (HTD_7_) and the *in vitro* and *in vivo* antileishmanial efficacy of an alkaloid 10-hydroxy chondrofoline. The results presented here indicated the potential antileishmanial use of an alkaloid obtained from a plant source, i.e., 10-hydroxy chondrofoline, and an antimalarial drug, i.e., tafenoquine against the *L. tropica* (HTD_7_). In vitro results indicated 10-hydroxy chondrofoline to be the most effective against the amastigotes of the parasite. Subsequent *in vivo* (using murine CL model) experiments using 10-hydroxy chondrofoline produced statistically significant results in decreasing the lesion size in comparison with the negative control. Overall, this study might help in the future in our quest to find safe and effective new compounds for the treatment of CL.

## Figures and Tables

**Figure 1 pharmaceuticals-15-01005-f001:**
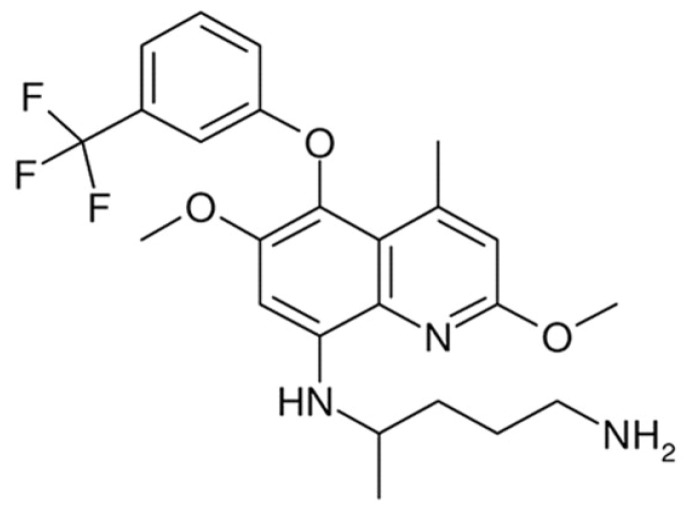
Structure of tafenoquine.

**Figure 2 pharmaceuticals-15-01005-f002:**
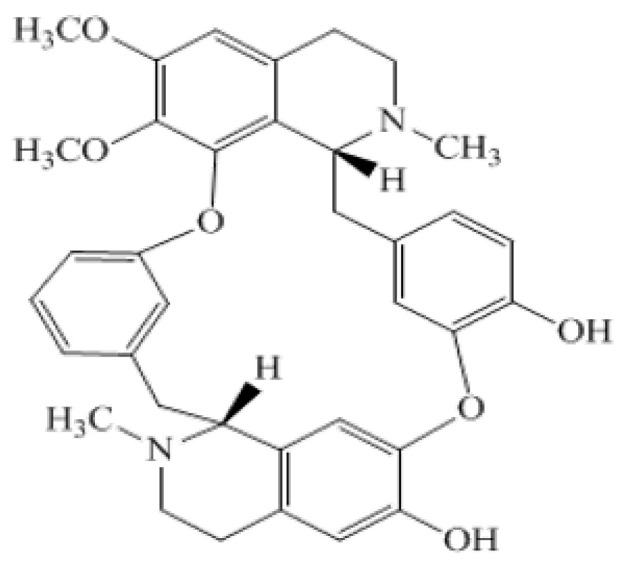
Structure of 10-hydroxy chondrofoline.

**Figure 3 pharmaceuticals-15-01005-f003:**
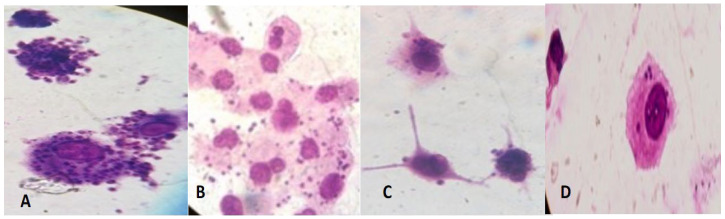
Giemsa stained THP-1 cells infected with *Leishmania tropica* 48 h post-infection and post-treatment with meglumine antimonate (**A**) Untreated control THP-1 (**B**) Treated with meglumine antimoniate (**C**) 10-hydroxy chondrofoline (**D**) Tafenoquine.

**Figure 4 pharmaceuticals-15-01005-f004:**
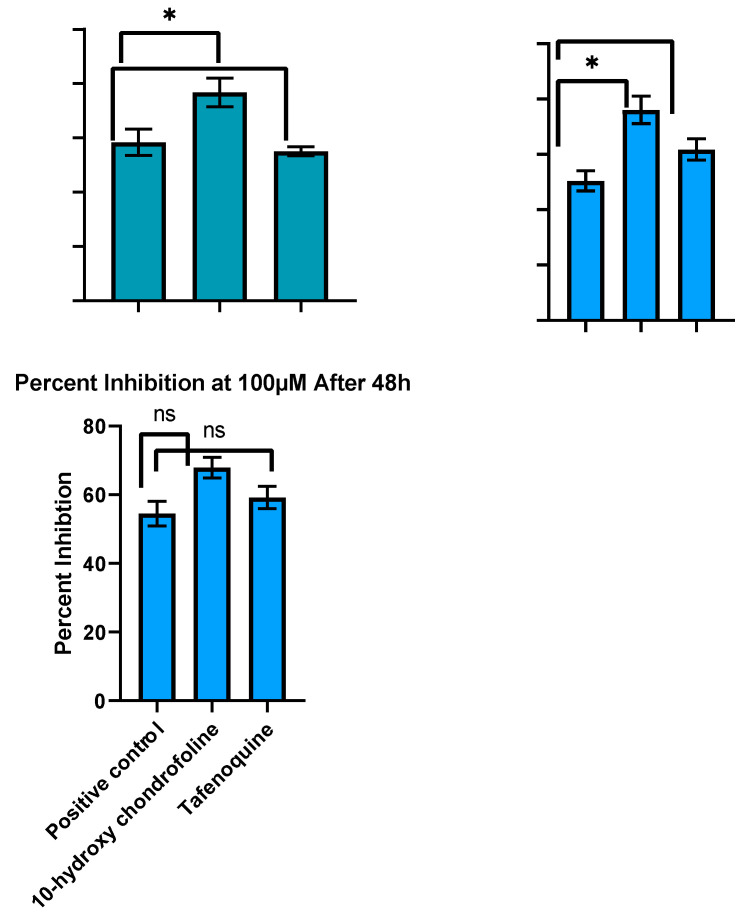
Percent inhibition (against amastigotes, *in vitro*) 48 h post-treatment with the test compounds at 25, 50, and 100 µM, respectively. Data are expressed as mean ± standard deviation (*n* = 3). * *p* > 0.05, ns = not significant.

**Figure 5 pharmaceuticals-15-01005-f005:**
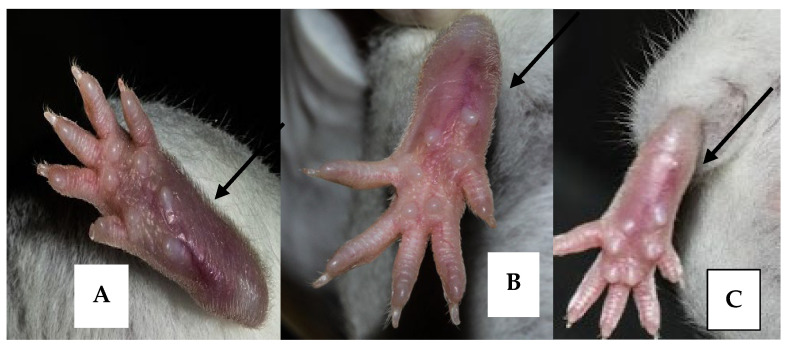
Representative images of the difference between the size of cutaneous lesions of the infected footpad after 20 days of (**A**) Negative control (Placebo) (**B**) 10-hydroxy chondrofoline (**C**) Positive control (meglumine antimoniate).

**Figure 6 pharmaceuticals-15-01005-f006:**
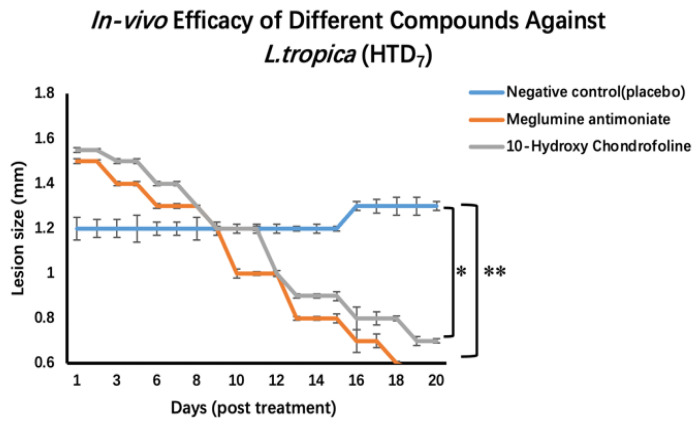
In vivo efficacy of 10-hydroxy chondrofoline in an experimental model of CL. Data are represented as mean ± standard deviation (*n* = 6). Statistically significant differences are given as ** represents *p* < 0.002 and * *p* = 0.002.

**Figure 7 pharmaceuticals-15-01005-f007:**
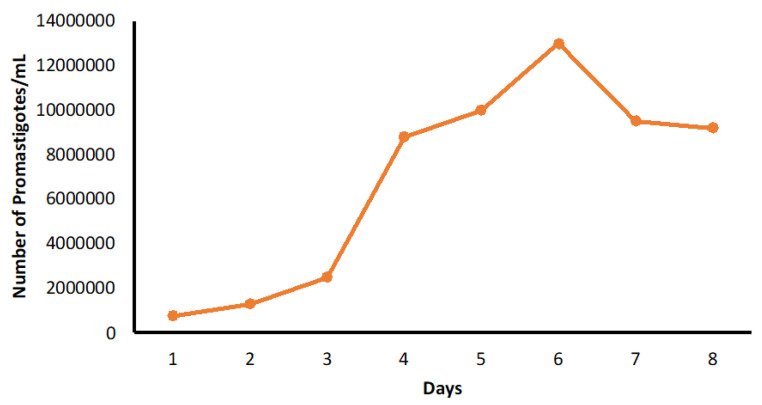
Growth curve for *L. tropica* HTD_7_ in culture.

**Table 1 pharmaceuticals-15-01005-t001:** In vitro cytotoxicity of the compounds and the standard drug (100 µM) against THP-1 macrophages after 48 h.

Compound	Mean Percent Viability ± SD
Control THP-1 cells only	97.1% ± 1.1
Meglumine antimonate + THP-1 cells	94.84% ± 2.2
10-Hydroxy Choline + THP-1 cells	96.62% ± 1.5
Tafenoquine + THP-1 cells	97.02% ± 1.4

## Data Availability

Data is contained within the article.

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
