# Peer review of "Efficacy Evaluation of 10-Hydroxy Chondrofoline and Tafenoquine against Leishmania tropica (HTD7)"

_pharmaceuticals, 2022, doi:10.3390/ph15081005_

Round 1

Author Response

Efficac Evaluation of 10-Hydroxy Chondrofoline and Tafenoquine Against Leishmania

tropica (HTD7)

REVIEWER 1

The manuscript by Shah et al. describes the testing of a possible anti-leishmanial compound, 10-

hydroxy chondrofoline, against intracellular amastigotes of Leishmania tropica and against L.

tropica-causes footpad lesions in BALB/c mice. The authors find that their test compound and their

control compounds show little toxicity against the macrophage-like cell line THP1, but reduce in

vitro infections of L. tropica in THP1 cells up to 90%, albeit at a very high dose of 40-160 μM.

Compared with meglumine antimonate and tafenoquine, the tested 10-hydroxy chondrofoline

exhibits slightly better efficacy in vitro. Preliminary in vivo testing in BALB/c mice shows

comparable efficacy of meglumine antimonate and the test compound.

The results must be documented much better; the Methods are woefully incomplete. The animal

testing requires additional biological samples, and the authors should use molar dimensions for

concentrations.

Specific points:

  1. line 74: refer precisely to the parasite species, i.e. T. brucei, as "Trypanosomiasis" can mean

Chagas disease or HAT in this context, and both are very different!

Thank you for the kind suggestion-we have modified this

  1. lines 121ff: the compounds used are very divers. Concentrations given in μg/mL are therefore

useless. Always give concentrations in molar dimensions.

We thank you for your kind input-as per the suggestion, we have repeated the in vitro experiments at 3 concentrations, i.e., 25, 50 and 100 μM and the results have been appropriately reported.

  1. line 135: the inhibiting concentration 50% does not signify killing! That would be LD50 (lethal

dose 50%), which cannot be determined in the experimental setup.

Thank you for the kind input- we have rectified the mistake as per the suggestion

  1. lines 130ff: again effective concentrations should be given as μM, since the different molecular

masses result in different molarities, confusing the results.

Thank you, this has been addressed now

  1. lines 161ff: please explain the groups! Are those different biological repeats?

Thank you for highlighting this- the groups have been already explained in the method section (4.2.5), where we have indicated how many groups were used as well as the number of test animals used per group. Briefly, we used three groups, i.e., positive control, negative control, and the test group (for 10-hydroxy chondrofoline). Each group consisted of 6 mice (a total of 18) and the data obtained was from these 6 mice (n=6) per group.

  1. section 4.2.4: this is insufficient. This method section lacks crucial information:
  2. growth phase of the parasites,
  3. multiplicity of infection,
  4. culture vessels used for the experiment,
  5. incubation time for infection
  6. incubation time for parasite persistence
  7. fixation and detection method
  8. methods of quantification for percentage of infected macrophages and mean parasite

load

  1. number of macrophages evaluated
  2. number of true biological repeats
  3. measures against observer bias

We are thankful to the respected reviewer for highlighting this point. This section has been rewritten again by keeping in view the valuable suggestions. We have now included the required information, which includes a detailed description of how this experiment was carried out. Moreover, a growth curve was also added (Figure 7) in section 4.2.1

  1. At least 2 biological repeats should be shown to demonstrate in vivo efficacy.

Thank you for the suggestion. The in vivo experiment was carried out as per the established protocol, which included both positive e and negative control, as well as the number of animals used per group, was 6. Therefore, we think that the set-up of the experiment was according to the established protocols.

  1. Information regarding the ethical clearance for the animal experiments is lacking.

Thank you- this has been added in section 4.2.5

The authors are thankful to the respected reviewer for the valuable suggestions which have certainly improved the quality of the manuscript.

Reviewer 2 Report

The manuscript “Efficacy Evaluation of 10-Hydroxy Chondrofoline and Tafenoquine Against Leishmania tropica (HTD7)” present data concerning the assays of 10-hydroxy chondrofoline and tafenoquine for their antileishmanial activity against intracellular (inside macrophages) amastigotes of L. tropica (HTD7). The data were reported to meglumine antimonite as positive control. My overall comment is that the data presented interest considering the need to find new drugs for this neglected disease able to overcome the secondary effects. Hence, the study selected natural products and the assays against THP-1 cells demonstrated their lack of toxicity. The efficiency was moreover demonstrated by the in vivo activity in murine cutaneous leishmaniasis (CL) model and data indicated that 10-hydroxy chondrofoline is a very potent species. The experiment were correct conducted and the approval of Ethical committee is presented. The data are proper discussed and in vivo experiments conducted on murine CL model produced statistically significant results in decreasing the lesion size in comparison with the negative control. As result, this study are important in order to find safe and effective new species for the treatment of CL.

I therefore recommend minor revision having in view the following aspects:

-        10-Hydroxy Chondrofoline must be corrected as 10-hydroxy chondrofoline together with other words provided with uppercase letters inside of the text.

-        The IC50 must be corrected as IC50 in whole paper.

-        The full name of some abbreviations must be provided first when appear in text even these are well known by the scientific community (i.e. THP-1, ATP)

-        In vitro at 2.1. must be provided in Italic style.

-        The References must be provided according with the Journal specifications.

Author Response

REVIEWER 2

The manuscript “Efficacy Evaluation of 10-Hydroxy Chondrofoline and Tafenoquine Against Leishmania tropica (HTD7)” present data concerning the assays of 10-hydroxy chondrofoline and tafenoquine for their antileishmanial activity against intracellular (inside macrophages) amastigotes of L. tropica (HTD7). The data were reported to meglumine antimonite as positive control. My overall comment is that the data presented interest considering the need to find new drugs for this neglected disease able to overcome the secondary effects. Hence, the study selected natural products and the assays against THP-1 cells demonstrated their lack of toxicity. The efficiency was moreover demonstrated by the in vivo activity in murine cutaneous leishmaniasis (CL) model and data indicated that 10-hydroxy chondrofoline is a very potent species. The experiment were correct conducted and the approval of Ethical committee is presented. The data are proper discussed and in vivo experiments conducted on murine CL model produced statistically significant results in decreasing the lesion size in comparison with the negative control. As result, this study are important in order to find safe and effective new species for the treatment of CL.

I therefore recommend minor revision having in view the following aspects:

-        10-Hydroxy Chondrofoline must be corrected as 10-hydroxy chondrofoline together with other words provided with uppercase letters inside of the text.

Thank you for the kind suggestion- this has been addressed in the whole manuscript.

-        The IC50 must be corrected as IC50 in whole paper.

Thank you for the kind suggestion- this has been addressed in the whole manuscript.

-        The full name of some abbreviations must be provided first when appear in text even these are well known by the scientific community (i.e. THP-1, ATP)

Thank you for the kind suggestion- the abbreviations have been added.

-        In vitro at 2.1. must be provided in Italic style.

Thank you for the kind suggestion- this has been addressed

-        The References must be provided according with the Journal specifications.

Thank you for the kind suggestion- this has been addressed as per the journal guidelines

The authors are thankful to the respected reviewer for the valuable suggestions which have certainly improved the quality of the manuscript.
